# Whole blood DNA methylation analysis reveals respiratory environmental traits involved in COVID-19 severity following SARS-CoV-2 infection

Guillermo Barturen [1] ✉, Elena Carnero-Montoro[1], Manuel Martínez-Bueno[1], Silvia Rojo-Rello[2], Beatriz Sobrino[3,4], Óscar Porras-Perales [3,4], Clara Alcántara-Domínguez [5], David Bernardo[6,7] & Marta E. Alarcón-Riquelme [1,8] ✉

SARS-CoV-2 infection can cause an inflammatory syndrome (COVID-19) leading, in many cases, to bilateral pneumonia, severe dyspnea, and in ~5% of these, death. DNA methylation is known to play an important role in the regulation of the immune processes behind COVID-19 progression, however it has not been studied in depth. In this study, we aim to evaluate the implication of DNA methylation in COVID-19 progression by means of a genome-wide DNA methylation analysis combined with DNA genotyping. The results reveal the existence of epigenomic regulation of functional pathways associated with COVID-19 progression and mediated by genetic loci. We find an environmental trait-related signature that discriminates mild from severe cases and regulates, among other cytokines, IL-6 expression via the transcription factor CEBP. The analyses suggest that an interaction between environmental contribution, genetics, and epigenetics might be playing a role in triggering the cytokine storm described in the most severe cases.

SARS-CoV-2 virus infection has affected millions of people during the last years worldwide. Most infected SARS-CoV-2 individuals remain asymptomatic or with mild symptoms that do not require hospitalization (~81%), while in others, the virus causes a disease called COVID-19 that primarily affects the lungs leading, in many cases, to bilateral pneumonia, severe dyspnea and in ~5% of the infected individuals, death[1,2].

Several genetics, transcriptomics, and proteomics molecular studies have been performed to date, disentangling important pathogenic molecular mechanisms of the disease[3–13]. In summary, SARS-CoV-2 infects the cells expressing surface receptors ACE2 and TMPRSS2[6] causing cell damage due to its replication and release from the host cell. This process triggers in the surrounding cells the production of pro-inflammatory cytokines and chemokines

[1]GENYO. Center for Genomics and Oncological Research Pfizer/University of Granada/Andalusian Regional Government, Granada, Spain. [2]Servicio de Microbiología e Inmunología, Hospital Clínico Universitario de Valladolid, Valladolid, Spain. [3]Servicio de Enfermedades Infecciosas, Hospital Regional de Málaga, Málaga, Spain. [4]IBIMA. Instituto de Investigación Biomédica de Málaga, Málaga, Spain. [5]Lorgen G.P., S.L., Business Innovation Center - BIC/CEEL, Technological Area of Health Science, Granada, Spain. [6]Mucosal Immunology Lab. Unidad de Excelencia Instituto de Biomedicina y Genética Molecular de Valladolid (IBGM, Universidad de Valladolid-CSIC), Valladolid, Spain. [7]Centro de Investigaciones Biomédicas en Red en Enfermedades Infecciosas (CIBER-INFEC), Madrid, Spain. [8]Unit of Inflammatory Chronic Diseases, Institute of Environmental Medicine, Karolinska Institutet, Stockholm, Sweden. ✉e-mail: guillermo.barturen@genyo.es; marta.alarcon@genyo.es

(including IL-1, IL-6, IL-8, IL-10, TNF and interferon inducible molecules, among others), which establishes a pro-inflammatory response mediated by the accumulation of specific immune cells[4]. In severe cases, an overproduction of cytokines is observed in lung tissues, known as cytokine storm, thus provoking an over-response of the immune system and causing tissue damage. In the most critical cases, the cytokine storm spreads to other organs leading to multi-organ failure and death. Currently, the molecular mechanisms and the pathophysiology behind COVID-19 progression are largely studied and well established, but it is still unclear what makes some individuals develop the severe illness. In this sense, underlying genetic variation[7,8,10] and the presence of various comorbidities have been identified as risk factors, such as diabetes, obesity, hypertension, chronic lung disease or even neurological disorders[14,15]. Also, life style habits that might be causing the previous conditions have been also related to COVID-19 illness as smoking, as well as age, sex or ethnicity[16,17]. However, it is unclear how these comorbidities, environmental and demographic conditions together with genetics, predispose and regulate the molecular mechanisms behind COVID-19 severity.

In order to shed light into the molecular relationship between risk factors and the regulation of the mechanisms behind the COVID-19 severity, here we present a DNA methylation EWAS (epigenome wide association analysis) combined with DNA genotyping for 473 and 101 SARS-CoV-2 lab positive and negative tested individuals, respectively, recruited in two independent clinical centers. In addition to the study of the epigenetic regulation of COVID-19 pathogenic mechanisms, the DNA methylation changes associated with COVID-19 progression, and their genetic regulation were put in context by comparing the results with DNA methylation changes occurring in systemic autoimmune diseases (SADs), and with GWAS (genome wide association analysis) and EWAS catalogues that collect multiple traits described as potential COVID-19 severity risk factors.

## Results

### COVID-19 severity is associated with impaired blood cell proportions and epigenetic activation of the innate immune response

Main blood cell type proportions were deconvoluted from the methylomes, showing a significant increase in neutrophil proportions associated with severity of the disease (Fig. 1a and Supplementary Fig. 1A, B). This imbalanced neutrophil proportion has been already shown in COVID-19 severity progression, and has been proposed as an early prognostic signature[1]. Besides cell proportion differences, significant differences in age and sex between groups were found in the discovery dataset (Wilcoxon test $p$-value < 0.05 for age in severe group compared to mild and negative individuals, and Fisher's exact test $p$-value < 0.05 for sex proportion in severe group compared with mild group). Methylation plates did not show batch bias, instead the largest bias observed was between cohorts and therefore were analyzed separately (Supplementary Fig. 1B, D). Based on these results, differential methylation analyses included as covariates: sex, age and the six major deconvoluted cell proportions.

Differential analyses were performed by pairs and longitudinally, after translating groups' severity into a numerical scale (severity analysis, hereafter). We identified 530 CpGs differentially methylated in at least one regression model, and confirmed in the replication cohort. Out of these, 43 DMCs were found in the severe-negative comparison, 347 in the mild-negative, 20 in severe-mild and 257 in the severity analysis (significant DMCs can be consulted in the Supplementary Data 1). We observed a high degree of sharing between DMCs obtained in different comparisons (Fig. 1b), except for the severe-mild DMCs which did not overlap with any of the other analysis results. These specific DMCs from the severe-mild analysis were hypermethylated in the severe condition. Overall, 24 DMCs, annotated into 17 different

genes were shared between severe-negative, mild-negative and with the severity analyses (Fig. 1b, c), which give a general idea of the epigenetic contribution to the progression of COVID-19. Most of the shared signatures are related to the activation of the viral defense type I interferon inducible genes (*OAS1-OAS2* hypermethylated and *PARP9-DTX3L, IFIT3, IRF7, TRIM22, MX1* hypomethylated), the hyperactivation of B and T lymphocytes (*CD38, EPSTI1, LAT* hypomethylated), and others, such as *EDC3*, known to interact with ACE2[18].

The influence of comorbidities on the results was tested by adding all comorbidity categories with a Fisher's exact test $p$-value < 0.05 (between at least two groups either in the discovery or the replication cohort) in the linear models. These were asthma, chronic heart disease, hypertension and current smokers out of 14 tested. All DMCs remained significant at a $p$-value below 5e-06 in the meta-analysis. The statistics for both discovery and replication models as well as for the meta-analysis showed a high correlation with an R-squared correlation ~ 1 and a $p$-value below 2.2e-16 (see Supplementary Fig. 1E–G).

DMCs localization enrichment analysis showed that hypermethylated changes related to SARS-CoV-2 infection are more prone to occur outside CGIs, particularly in introns. For the hypomethylated sites, these occur in enhancers (Supplementary Fig. 2A). These genomic regions are known to be hot-spots of DNA methylation changes[19]. However, most of the DMCs found in these analyses colocalize around the TSS (Transcription Start Site) and/or in the 5′-UTR of the nearest gene (Supplementary Fig. 2B), due to the EPIC array probe selection. This probes' preferential location facilitates the interpretation of the results, as hypermethylation and hypomethylation in 5′-end regions of the genes are mostly related to the inactivation and activation of gene expression, respectively[20,21].

### COVID-19 disease DNA methylation changes in neutrophils, B-lymphocytes and CD8+ T-lymphocytes regulate functional pathways related with autoimmune diseases and viral defenses

Functional enrichment analyses based on Reactome pathway database were performed taking into consideration the groups compared and the direction of the effects. An enrichment of hypomethylated signals at interferon-inducible genes, herein called IFN signature, and enrichment of hypermethylated signals at genes involved in FCGR phagocytosis and CD209 signaling (DC-SIGN) was observed when positive SARS-CoV-2 were compared to negative SARS-CoV-2 individuals (Fig. 2a). These pathways were also enriched in a probe-oriented enrichment pathway analysis, which considers known biases in EWAS array-based technologies[22] (Supplementary Fig. 3). The activation of IFN signature genes is related with an active viral infection and in particular with SARS-Cov-2 infection[9]. However, at DNA methylation level the impaired interferon response between mild and severe cases found at the transcriptional level[5] could not be observed (Supplementary Fig. 4). This suggests that exhaustion of the interferon signature might be controlled at a different regulatory level.

We performed interaction analysis between deconvoluted cell proportions and severity groups to identify which blood cell types are contributing to the epigenetic signatures. Our results suggest that interferon associated hypomethylation changes were mainly due to neutrophils and CD8+ T-lymphocytes (Fig. 2b), while hypermethylation changes were primarily occurring in B-lymphocytes. (Fig. 2b). This in turn, might be related to the inactivation of CD209 signaling (Fig. 2a). CD8+ T-lymphocytes also showed a number of significant hypermethylated interactions (Fig. 2b) that may be related with the inactivation of FCGR3A phagocytosis-related genes in these cells (Fig. 2a). Lastly, in the severe-mild analysis, methylation changes of the PIP3 activated AKT signaling pathway differentiated severe from mild COVID-19 patients (Fig. 2a). Genes related with this pathway were hypermethylated in severe cases compared with mild COVID-19 cases, being CD8+ T-lymphocytes the major contributors to these changes (Fig. 2b).

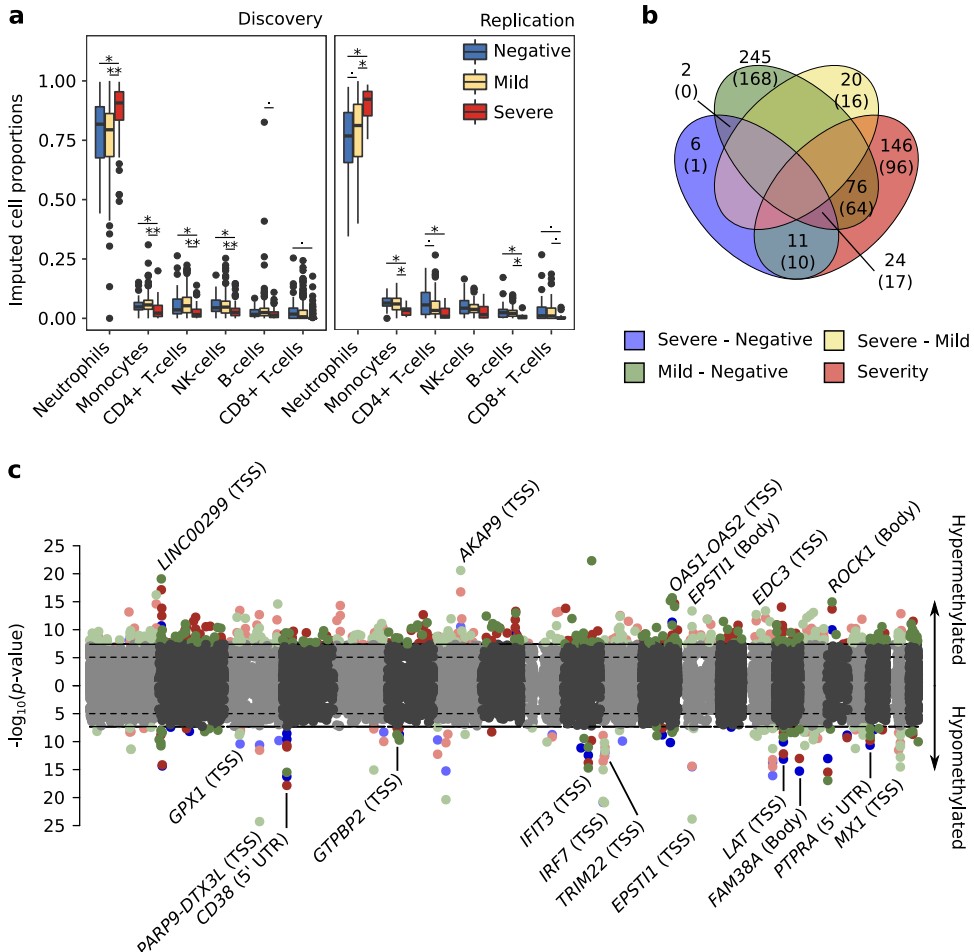

**Fig. 1 | COVID19 severity correlates with an increase in blood neutrophil proportion and epigenetic changes in genes related with the innate immune response. a** Methylome deconvoluted blood cell proportions are plotted by cohort (left panel discovery, right panel replication) and group (blue, 47 and 54 negative SARS-CoV2 lab tested individuals for discovery and replication; yellow, 269 and 91 positive individuals with mild symptoms for discovery and replication and red, 98 and 15 positive individuals with severe symptoms for discovery and validation). Paired differences were assessed by means of linear regression analysis (age and sex were included as covariates) and significance values plotted by pairs (·*p*-value < 0.05, **p*-value < 0.01 and ***p*-value < 1e-5). The center line denotes the median value, the box contains 25th to 75th percentiles of the dataset and the whiskers extend up to ± 1.5*IQR. **b** Venn diagram with the number of significant shared DMCs across the differential analysis performed (the number of annotated genes are included in parentheses). **c** Combined manhattan plots are shown for the differential analysis that share DMCs, hypermethylated and hypomethylated DMCs are divided into upper and lower side of the manhattan plot respectively. Genes annotated for the shared DMCs are depicted, including, in parentheses their co-localization with the annotated gene (TSS, Transcription Start Site: Body, gene body). Severe vs negative (blue), mild vs negative (green), severe vs mild (yellow) and pseudotime longitudinal analysis (red).

In order to validate the activation or inactivation of the enriched pathways as revealed by the DNA methylation changes, Reactome pathways' activity was estimated based on single-cell RNA-Seq information from publicly available analyses[11,12]. The analysis was focused on the cell-types that mostly contribute to the DNA methylation changes: CD8+ T-lymphocytes, B-lymphocytes and neutrophils, as revealed from the interaction results (Fig. 2b). In general, molecular pathway activities follow the DNA methylation changes at early sampling time points, which corresponds with our recruited cohorts. That is, the pathways that show hypomethylation in certain group(s) of individuals coincide with a higher transcriptome activity compared with the hypermethylated groups, at least in the cell-types in which the change has been predicted to occur (Supplementary Fig. 5). For example, the FCGR3A phagocytosis pathway activity is decreased with the severity of the disease in CD8+ T-lymphocytes, while the interferon signaling activity is increased with severity. Certainly, at the transcriptome level, the interferon exhaustion signature associated with severe cases, not previously seen at the DNA methylation level (Supplementary Fig. 4), can be appreciated for B-lymphocytes and CD8+ T-lymphocytes.

Finally, enrichment analyses were performed to assess to which other phenotypes or diseases the COVID-19 DMCs can be associated. For that, we used the information gathered in the EWAS Atlas catalog[23]. Except for severe-mild DMCs, the other 3 comparisons showed DNA methylation changes in CpGs that were previously associated with different autoimmune conditions, allergic conditions, and an asthma related trait (as fractional exhaled nitric oxide test), but also with differential respiratory related environmental exposures (air pollution and polybrominated biphenyl exposure) and/or comorbidities that reflect lifestyle habits such as body mass index, smoking or alcohol consumption (Fig. 2c).

**Respiratory environmental related epigenetic changes differentiate severe and mild COVID-19 patients and mild COVID-19 cases from systemic autoimmune disorders**

Significant DMCs from all the differential analyses performed were clustered together based on their methylation profile grouped by COVID-19 severity and divided into the two recruited cohorts (Fig. 3a). Hierarchical clustering reveals that aside from the significant values obtained in the linear regression models, not all trends of DMCs

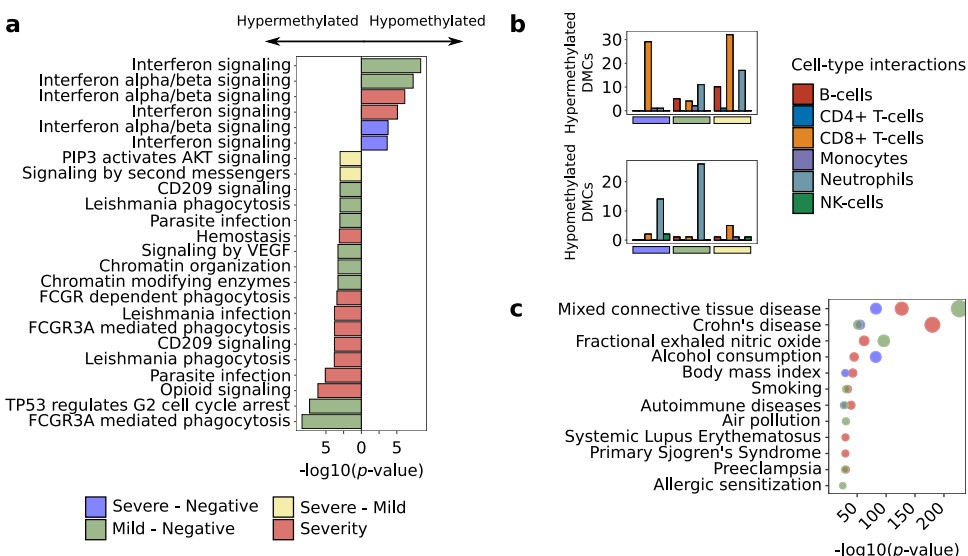

**Fig. 2 | COVID19 DNA methylation changes regulate autoimmune related functional pathways and associate with environmental respiratory related traits. a** Top 10 significant reactome database pathways (two-tailed hypergeometric *p*-value < 0.01) are shown by differential analysis. **b** Number of DMCs with significant interactions for each deconvoluted cell-type proportion (red, B-cells; blue, CD4 + T-cells; orange, CD8 + T-cells; purple, monocytes; blue, neutrophils and green, NK-cells) are split into hypermethylated (upper panels) and hypomethylated (lower panel) and divided into the differential analysis. **c** EWAS traits enrichments (two-tailed hypergeometric *p*-value < 1e-10) for each differential analysis are shown (MethBank database). Severe vs negative (blue), mild vs negative (green), severe vs mild (yellow) and pseudotime longitudinal analysis (red).

methylation changes are exactly replicated in both cohorts. Thus, 4 DMC modules were obtained based on the hierarchical clustering where DNA methylation changes were stable: S.Ho, composed by CpGs with a hypomethylation profile along COVID-19 severity; S.He, characterized by a hypermethylation profile along COVID-19 severity; M.Ho, in which hypomethylation events are observed in mild as compared with severe cases; and M.He, in which hypermethylation occurs in mild as compared with severe cases. In order to give further robustness to the results, CpGs reliability within modules was assessed by means of the reliability metric defined by Sugden et al.[24] and the log2 fold-changes between groups were compared with a recently published dataset[25]. The reliability metric of the CpGs within modules S.Ho, S.He and M.He were significantly higher than the overall CpGs reliability (2.3e-10, 3.4e-04 and 1.3e-12 Kolgomorov-Smirnov *p*-values, respectively (Supplementary Fig. 6A). And the log2 fold-changes were replicated in the external cohort (positive correlation *p*-values below 1e-05, Supplementary Fig. 6B–D). However, the M.Ho module showed low reliability values and the methylation changes were not replicated in the external cohort. Thus, this module was discarded from further analyses.

In summary, Reactome pathway enrichment analysis on the 3 modules (Fig. 3b, Supplementary Fig. 7) replicated the previous enrichments found for the DMCs grouped in the linear regression analysis (Fig. 2a). Interestingly, a new additional pathway appeared to be enriched in the S.He module, related with potential therapeutics for SARS, which suggests that several of the proposed therapeutic targets for SARS infection are based on the activation of hypermethylated molecular pathways during the course of the COVID-19 disease.

On the other hand, EWAS Atlas catalog enrichments were performed by modules, revealing that autoimmune and asthma related traits were mostly enriched in S.Ho and S.He modules, while the differential respiratory environmental related traits were enriched in the M.He module (Fig. 3c). The M.He module is hypermethylated in mild COVID-19 cases as compared with severe cases and negative controls, suggesting that differential respiratory environmental exposures might play a protective role against severe COVID-19 progression, upon SARS-CoV-2 infection.

TFBS motif analysis reveals specific TFBS motifs enriched for the different modules (Fig. 3d). S.Ho module was mainly enriched in interferon regulatory TFBSs, in line with the Reactome pathway enrichment results. Among the other results, the enrichment of the CEBP motif in the M.He module stands out. CEBP is a transcription factor related with the inflammatory immune response through its cooperation with IL-6, and stimulating the transcription of different pro-inflammatory cytokines[26].

Given the potential relationship between the COVID-19 affected molecular pathways and autoimmune disorders, DNA methylation profiles were compared between COVID-19 and the systemic autoimmune disease PRECISESADS collection[27], which includes DNA methylation information from seven SADs (Fig. 3e). Both, severe and mild related DNA methylation changes correlated with systemic autoimmune disorders for S.He module, having a slightly higher intensity in severe COVID-19 patients. S.Ho module correlations were also significantly positive, except for the RA and SSc comparison with mild cases, which presented no significant correlations. In general, contrary to SLE and pSjS, RA and SSc patients do not express the IFN signature enriched in S.Ho module[28]. Thus, this result might be related with the presence of two signatures contributing to this module, one related with the interferon, which highly correlates with most interferon related SADs, and another one that correlates between severe COVID-19, RA and SSc. In order to further investigate the differential correlation between SADs in this particular module, the strongest interferon-related hypomethylated CpGs found in SADs and COVID patients (logFC < −0.25) were removed from the correlation analyses (annotated in *TRIM22-TRIM5*, *PARP9-DTXL3*, *RUNX1*, *IFIT3*, *IRF7*, *EPSTI1*, *MX1* and *ADAR* genes). The resulting correlation after discarding these CpGs showed a dramatic reduction in interferon related SADs, while correlations of severe cases with RA and SSc were preserved (Supplementary Fig. 8). This suggests that the remaining CpGs (annotated in genes such as *CCDC61*, *CD38*, *FAM38A*, *LAT*, *TREX1* or *NFAT5*, among others) differentially contribute to similarities between COVID-19 progression and SADs, some of them regulating the activation and differentiation of T and B lymphocytes. On the other hand, M.He module showed a strong correlation for severe, and a strong anti-correlation with mild cases, thus differentiating mild cases from SADs.

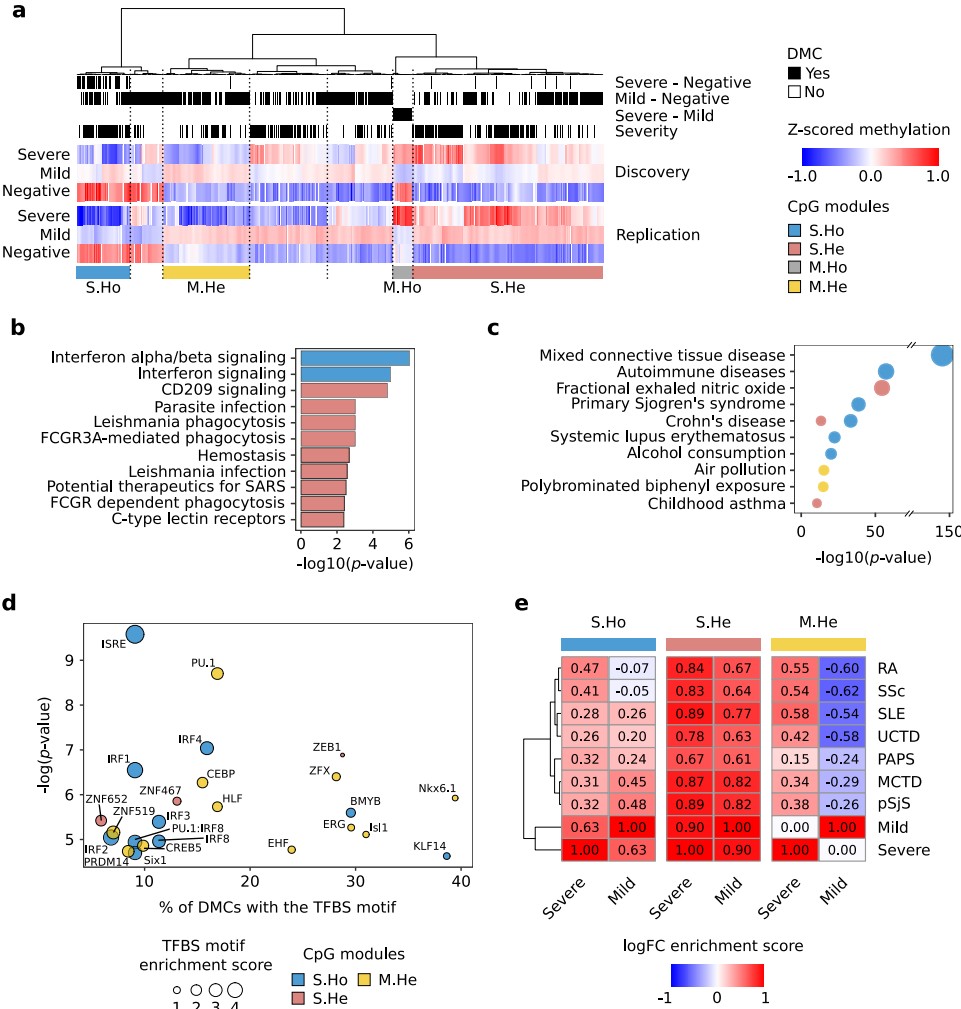

**Fig. 3 | Epigenetic changes in CpGs associated with environmental respiratory traits differentiate COVID19 progression and mild cases from autoimmune disorders. a** Hierarchical clustering of methylation DMCs for both discovery and replication cohorts (Ward's hierarchical agglomerative clustering with Pearson correlation as distance is used). Individual methylation values are averaged by severity from severe cases (top), mild cases (middle) to negative lab tested SARS-CoV2 (bottom). The annotations in the upper part of the plot refer to the analysis to which each CpG is differentially methylated (black). Four CpG modules highly replicated between cohorts, were selected from the hierarchical clustering: S.Ho (blue, hypomethylated with the severity), S.He (red, hypermethylated with the severity) and M.He (yellow, hypermethylated in mild compared with severe

patients and healthy controls). **b** Reactome significant pathways by CpG module (two-tailed hypergeometric *p*-value < 0.01) are shown. **c** MethBank EWAS trait enrichment by CpG module (two-tailed hypergeometric *p*-value < 1e-10) are shown. **d** Significant overrepresentation of transcription factor binding site prediction (HOMER, two-tailed hypergeometric *p*-value < 0.001) is depicted by CpG module. **e** Average log2FC Pearson correlations between COVID19 severity groups and seven different systemic autoimmune conditions (SLE Systemic lupus erythematosus, RA Rheumatoid arthritis, pSjS Primary Sjögren's syndrome, SSc Systemic sclerosis, MCTD Mixed connective tissue disease, PAPs Primary antiphospholipid syndrome, UCTD Undifferentiated connective tissue disease). DMCs are grouped by CpG modules.

## Specific hypermethylation in mild cases shows a minor genetic contribution, while meQTLs are enriched in SNPs associated with environmental traits

As the specific mild hypermethylated changes (M.He) were mainly associated with environmental traits, we next interrogated whether there is genetic contribution behind these epigenetic changes, and how genetics contribute to the DNA methylation modules. In this sense, DNA methylation heritability was calculated for each CpG in the modules. Two independent methods showed high agreement in heritability estimation (Supplementary Fig. 9A), so for the subsequent analysis, the variance decomposition model was selected. Genetic contribution to methylation variability was shown to contribute differentially between modules, being larger in S.Ho and S.He than in the M.He module (Fig. 4a). This is in agreement with the larger environmental contribution to M.He shown by EWAS trait enrichments. Additionally, covariates such as SARS-CoV-2 infection, age, and sex, did not modify the genetic contribution to the DNA methylation changes

(Supplementary Fig. 9B). On the contrary, S.Ho and S.He modules were significantly modified by SARS-CoV-2 infection, while M.He variation might be driven by other covariates or environmental factors that, unfortunately, were not recorded in these cohorts (Supplementary Fig. 9C).

In order to investigate deeper into the genetic contribution on the DNA methylation changes observed during COVID-19 progression, cis-meQTLs (methylation quantitative trait loci) were assessed (significant results can be consulted in Supplementary Data 1). Linear regression models were independently fit for each group (FDR < 0.05 for at least one group), showing that nearly 50% of the CpGs in each module were associated with at least one SNP (Supplementary Fig. 9D). In total, 7899 unique meQTLs were significant for at least one of the groups, composed of 7548 SNPs and 175 CpGs (out of 352 DMCs) with an average of 45 ± 84 SNPs by CpG. This suggests that nearly half of the DNA methylation changes found are being regulated by large blocks of SNPs in cis. MeQTLs were classified according

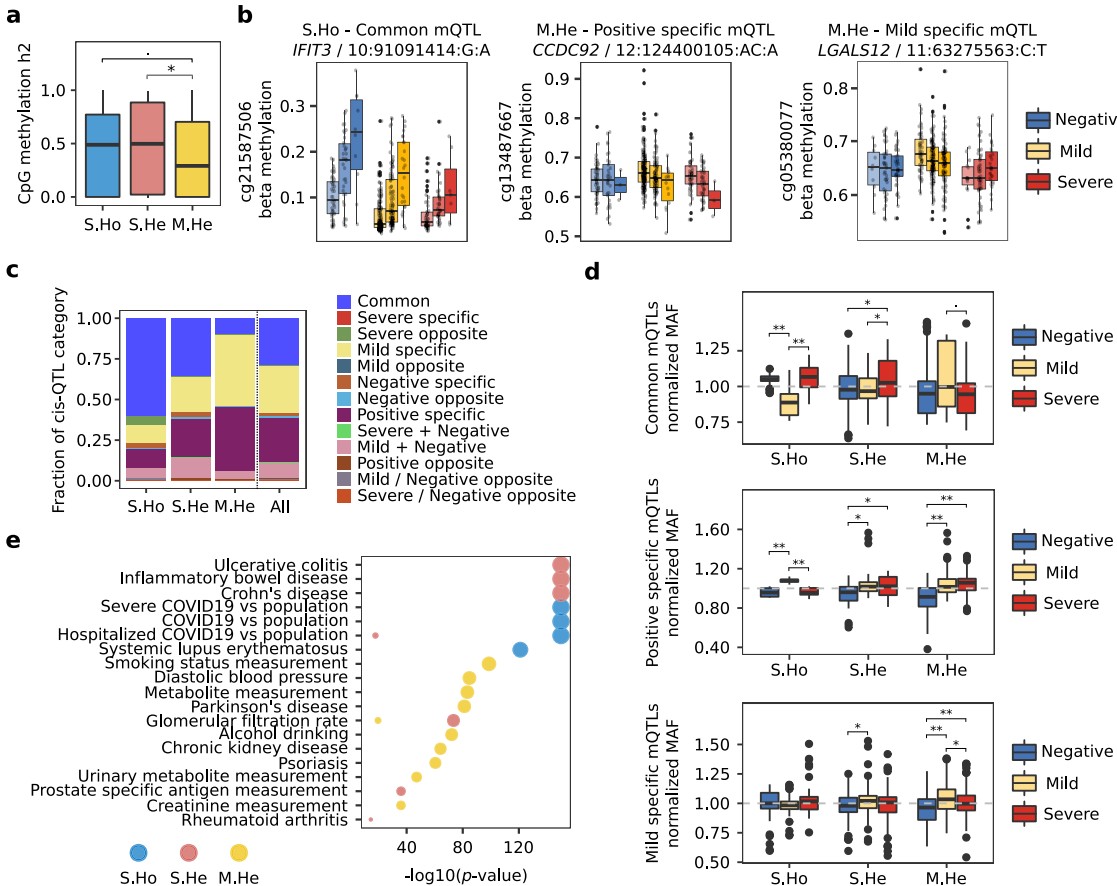

**Fig. 4 | Genetics contributes differentially to progressive and mild specific DNA methylation changes. a** Genetic contribution in terms of the fraction of the variance explained (heritability, h2) of individual CpG methylation changes is shown by DNA methylation module. Statistical differences are assessed by means of two-tailed Wilcoxon test *p*-values. **b** Three significant meQTLs regulating DNA methylation levels are shown divided by severity group and genotype. From left to right, a common meQTL for all three severity groups in the S.Ho module, a positive specific meQTL and a mild specific meQTL for M.He module are depicted. **c** Fraction of meQTL categories are plotted by module and for all significant DMCs together. **d** Normalized MAFs for the largest meQTL categories (common meQTLs, positive specific meQTLs and mild specific meQTLs) represented in at least three modules

(S.Ho, S.He and M.He) are shown divided by severity group. Two-tailed Wilcoxon test *p*-values were calculated between severity groups. **e** Enrichment of GWAS catalog and COVID-19 Host Genetics Initiative associated SNPs are shown by CpG module (two-tailed hypergeometric *p*-value < 1e-10). MeQTL analyses were performed on 101 negative SARS-CoV2 lab tested individuals, 360 positive individuals with mild symptoms and 113 positive individuals with severe symptoms. In the boxplots, the center line denotes the median value, the box contains 25th to 75th percentiles of the dataset and the whiskers extend up to ± 1.5*IQR. S.Ho module is depicted blue, S.He in red and M.He in yellow. Severe, mild and negative individual groups are colored in red, yellow and blue respectively.

to the significance of the SNP-CpG association by group, and labelled as follows: a meQTL was considered as mild-specific, when the significant association (*p*-value < 0.05) was only found in COVID-19 mild cases, or positive specific, when both mild and severe cases showed a significant association (Fig. 4b). MeQTLs classification showed a differential genetic regulation by module (Fig. 4c), where methylation changes following COVID-19 progression (S.Ho and S.He modules) were enriched in meQTLs shared by all the groups (common mQTLs). This suggests that the genetic regulation of these DNA methylation changes does not depend on the severity of the disease but are a general regulatory mechanism. On the other hand, meQTLs in the M.He module were mostly identified as group-specific meQTLs, with a large fraction of mild and positive specific ones. The genetic regulation specificity of M.He is also supported by significant differences in the normalized MAF (each group minor allele frequency divided by all groups' minor allele frequency) for the mild and positive specific meQTLs (Fig. 4d). MAFs showed a higher frequency in positive specific meQTLs in mild and severe groups as compared to negative individuals, while the mild specific meQTLs showed a higher MAF in mild cases. Surprisingly, MAF differences were found between mild as compared to severe and negative individuals for

common and positive specific meQTLs in S.Ho modules, which might indicate a differential genetic regulation also for mild individuals for the S.Ho signature (Fig. 4d).

The enrichment by module of the significant meQTLs was tested for SNPs previously known to be associated with different traits. In this sense, meQTLs trait enrichments were performed considering the GWAS catalog database[29] and the COVID-19 associated SNPs from the COVID-19 Host Genetics Initiative[7,8]. The results showed a strong enrichment of SNPs associated with COVID-19 and interferon related autoimmune diseases (systemic lupus erythematosus) in the meQTLs regulating the S.Ho module while SNPs associated with non-interferon related autoimmune diseases were observed in the S.He module (Fig. 4e). On the other hand, M.He meQTLs were enriched with environmental related SNPs (Fig. 4e), mimicking the enrichments shown above for the EWAS catalog. Interestingly, two different COVID-19 GWAS regions were regulating the S.Ho and S.He modules. In the case of the S.Ho module, its cis-meQTLs are composed of SNPs at the 3p21.31 GWAS peak, found to be associated with severe, hospitalized, and in general SARS-CoV-2 lab positive tested patients compared with the general population[7,8]. While the S.He module was enriched in SNPs located at the 8q24.13 GWAS peak, only found to be statistically

significant in hospitalized COVID-19 patients compared to the general population[8].

## Discussion

The EWAS of SARS-CoV-2 infection reveals the regulation by DNA methylation of important functional pathways related with COVID-19 progression. It also reveals specific epigenetic differences between severe and mild patients. Differentially methylated CpG sites were shared between severe and mild cases, mainly associated with the activation of interferon signaling pathway and the hyper-activation of B and T lymphocytes. These pathways have been previously associated with COVID-19 severity in transcriptome studies[9,30], showing in this study that the regulation of these pathways is being mediated by epigenetic changes at the promoter level of the implicated genes (Fig. 1).

In addition to the DMCs shared between the differential analyses, the pathway enrichment analysis for the individual regression models showed the epigenetic dysregulation of specific pathways such as CD209 signaling (DC-SIGN), the FCGR-mediated phagocytosis pathway and AKT signaling in specific blood cell-types (Fig. 2), however the latter was enriched in low reliable probes, and thus discarded from further analyses (Supplementary Fig. 6). CD209 is primarily expressed in dendritic cells and B-lymphocytes, and its interaction with CD209L, expressed in SARS-CoV-2 target tissue endothelial cells, has been described to facilitate the entry of the virus[31]. Thus, hypermethylation and the consequent under-expression of the CD209 signaling pathway might be playing a protective role during SARS-CoV-2 infection. Additionally, CD209 activation has been shown to promote B-lymphocyte survival[32]. However, this process does not seem to be occurring in SARS-CoV-2 infection as shown by the B-lymphocyte depletion observed in the deconvolution analysis (Fig. 1a). The FCGR phagocytosis pathway is involved in the antibody-antigen complex clearance and the antibody dependent cellular mediated cytotoxicity. CD8+ T-lymphocytes expressing FCGR3A (CD16) have been described to acquire natural killer (NK) cell-like functional properties, thus contributing to their cytotoxic functionality, increased for instance, in chronic hepatitis C virus infections[33]. Recently, suppression of cytotoxic activity has been described on CD8+ T-lymphocytes and NK-cells from severe COVID-19 patients[34], which in light of our DNA methylation results might be impaired, as could be explained by the DNA hypermethylation of genes of the FCGR3A phagocytosis pathway that we observe. Based on our results, these two pathways seem to be associated with the progression of the disease, showing significant DNA methylation changes along its course. Other important genes, not annotated in these pathways, were found to show methylation differences, as for example *EDC3*. Interestingly, hypermethylation of *EDC3* in severe cases might be mediating the overexpression of the ACE2 protein in SARS-CoV-2 patients, thus favoring infection[6]. *EDC3* is a component of a decapping complex that promotes removal of the monomethylguanosine (m7G) cap from mRNAs, being therefore an important protein during mRNA degradation. Its interaction with ACE2 has been experimentally validated and shown through STRING interaction network[18].

In addition to the COVID-19 EWAS results, considering that our cohort is barely below EWAS size standards[35], and in order to filter out potential false positive results, DMCs were grouped by hierarchical clustering and filtered by cohorts' similarity, reliability and replication with an external cohort (Fig. 3 and Supplementary Fig. 6). Three modules of co-regulated CpGs were found, where two of them were enriched in the functional pathways previously described. CD209 and FCGR phagocytosis pathways (S.He module) are hypermethylated with the severity of the disease, and both severe and mild cases, perfectly correlate with DNA methylation changes observed in SADs. Hypomethylation along COVID-19 severity (S.Ho module) was composed of

two signatures, an interferon related signature which correlates with interferon related systemic autoimmune diseases (as MCTD, SLE or pSjS) at both severe and mild cases, and a T and B lymphocyte activation signature, which correlates mainly with non-interferon related SADs (RA and SSc) for severe cases. The third module M.He, specifically hypermethylated in mild cases, is of particular interest. Severe DNA methylation changes as compared with negative controls were highly correlated with autoimmune conditions, while mild changes were negatively correlated. Additionally, and in contrast to the other CpG modules, the CpGs of M.He were not related with autoimmune but with respiratory environmental conditions. Further analyses on this module revealed an enrichment in transcription factor binding sites (CEBP, PU.1, ISL1 and CREB), which are known to positively regulate the levels of cytokines[26,36,37] related with COVID-19 severity[4] such as, IL-6, IL-1α, IL-1β, IL-12 and other pro-inflammatory cytokines containing a cAMP-responsive elements[38] (Fig. 3d). Interestingly, IL-1α has been proposed as an early marker of poor prognosis[4]. The CEBP transcription factor has an important role regulating IL-6 and IL-1β expression, whose elevated levels have been associated with severe complications of COVID-19 disease. The hypermethylation on M.He CpGs suggests a differential binding activity of these transcription factors in mild cases compared to the severe cases and the negative controls, in a module where DMCs are enriched in respiratory environmental traits. Altogether, our results suggest the existence of a relationship between environmental exposure and the protection against cytokine storm associated with the most critical outcomes of COVID-19 disease.

The genetic regulation of COVID-19 associated DNA methylation changes was also studied, finding important differences between modules (Fig. 4). In addition to a lesser genetic contribution to the DNA methylation changes in M.He module, the meQTLs associated to this module showed more group specificity than the S.Ho and S.He modules. Importantly, GWAS catalog enrichments for the meQTLs showed again a predominance of environmental traits related SNPs for the M.He module, which reinforces the idea of the importance of the environmental exposure during the regulation of its DNA methylation changes.

This study is an in depth EWAS comparing SARS-CoV-2 RT-PCR positive and negative individuals from a functional perspective. Previous EWAS had predictive purposes[25,39,40], having found in those studies a strong interferon signature which correlated with the progression of the disease and also discriminated between positive and negative SARS-CoV-2 individuals. In our results, this interferon-related signature showed an important epigenetic regulation of autoimmune-related functional pathways during COVID-19 progression that might differentiate severe from mild COVID-19 cases, as shown in previous EWAS. Some of these autoimmune-related pathways presented DNA methylation differences between severe and mild cases with lower genetic contribution, but with higher genetic specificity than changes that progress with the severity of the disease. Interestingly, these specific epigenetic changes were mainly related with environmental traits in terms of DNA methylation sites and the SNPs regulating these sites. Thus, in light of the results, the interaction between specific genetic variation and different environmental exposures or life habits might be dysregulating, via DNA methylation changes, autoimmune-related functional pathways which are, in turn, associated with worsening of SARS-CoV-2 infection. Despite the relationship between environmental exposure and COVID-19 severity suggested in previous epidemiological studies, this is the first time that this relationship is supported by genetic and epigenetic molecular information, thus, contributing to the understanding of the disease at the molecular level. Of special importance is the association of these environmental-related DNA methylation changes with the cytokine storm typical of the most severe COVID-19 cases.

**Table 1 | Cohorts' demographic and clinical information**

| | Discovery (10 Technical Batches) | | | | | Replication (3 Technical Batches) | | | | |
|---|---|---|---|---|---|---|---|---|---|---|
| | # | Age | Sex | Hospitalized | Deceased | # | Age | Sex | Hospitalized | Deceased |
| Negative | 47 | 63 ± 21 | 20 (43%) | — | — | 54 | 67 ± 20 | 27 (50%) | — | — |
| Mild | 269 | 67 ± 15* | 126 (47%) | 216 (80%)* | — | 91 | 61 ± 18* | 48 (53%) | 87 (96%)* | — |
| Severe | 98 | 76 ± 14* | 59 (47%) | 98 (100%) | 84 (86%) | 15 | 64 ± 18* | 7 (47%) | 15 (100%) | 10 (67%) |

#Number of individuals, age average ± standard deviation (Age), number and percentage of males (Sex), hospitalized individuals (Hospitalized) and deceased individuals (Deceased) are shown by severity group and cohort. *Discovery and replication cohorts showed significant differences in terms of age in mild and severe groups (two-tailed Mann-Whitney U test $p$-value < 0.05) and also in terms of numbers of hospitalized mild symptoms patients (two-tailed Fisher exact test < $p$-value 0.05).

## Methods

### Study design and cohorts

Whole blood samples from SARS-CoV-2 RT-PCR negative (101) and positive lab tested individuals (473) were obtained from two clinical centers (Hospital Clínico Universitario de Valladolid, discovery cohort and Hospital Regional Universitario de Málaga, replication cohort). Negative PCR individuals had no obvious evidence of infection at sampling and none of them were admitted to the hospital. The regional ethical committees from Andalucía (Comité Coordinador de Ética de la Investigación Biomédica de Andalucía) and from Valladolid (COMITÉ DE ÉTICA DE LA INVESTIGACIÓN CON MEDICAMENTOS ÁREA DE SALUD VALLADOLID) approved the protocols and gave their ethical approval for this study and all recruited individuals signed the informed consent prior to recruitment. Whole blood was sampled upon arrival to the emergency ward, within a week after first symptoms. Discovery and replication cohorts were recruited between March-April 2020 and August-October 2020, respectively. Individuals were classified based on the WHO clinical ordinal scale[41] (Supplementary Table 1): PCR negative individuals (uninfected, 0 scale), mild PCR positive individuals (ambulatory or hospitalized with mild symptoms, 1–4 scales), and severe PCR positive individuals (hospitalized with severe symptoms or died, 5–8 scales). The defined groups between cohorts were sex balanced, but slightly significant differences were found in terms of age (Table 1).

### Genomic analysis

**DNA extraction.** DNA was extracted from whole blood samples by means of the QIAamp DNA Blood Mini kit and the automatic platform QIAcube Connect. Afterwards, DNA quality was validated and normalized using the NanoDrop 2000c and the Qubit4.

**Genotyping.** DNA was normalized to 200–400 ng and genotyped with Illumina's Infinium GSA-24.v3.0 BeadChip (Illumina catalog number 20030771), following manufacturer's recommendations. Markers with genotyping rate > 99%, minor allele frequency > 1% and a $p$-value for Hardy-Weinberg Equilibrium > 1e-06 were selected. Samples showing genotyping rate < 98%, inconsistencies between reported and genetic sex and extreme heterozygosity values ($-0.2 <$ Fhet $< 0.2$) were eliminated. The kinship coefficient was calculated for each pair of samples and one member of each pair with a value >= 0.2 was removed. Based on a set of Ancestry Informative Markers (markers which maximize the allelic frequencies across 1000Genomes populations), individuals with non-European ancestry components were eliminated. The resulting dataset from this quality control process was imputed in the Michigan Imputation Server[42], using Minimac4 and 1000Genomes as reference panel[43]. After subsequent filtering of the imputation result we obtained a working dataset consisting of 504 samples and more than 9.5 million markers. Quality control of the genotyped data was performed with Plink2.0[44].

**Methylome profiling.** DNA methylation information was profiled with the Illumina's Infinium MethylationEPIC BeadChip (Illumina catalog number WG-317-1003), after sample normalization to 500 ng and bisulfite conversion with EZ-96 DNA Methylation Kit, as recommended by the manufacturer. Methylomes were quality controlled by genotype concordance (>= 0.8) using shared SNP probes between platforms (genotypes were extracted after imputation but without post filtering), sex prediction agreement (outliers > 5 standard deviations), signal from noise detection $p$-value < 0.1 and minimum number of beads (>3) that passed the detection $p$-value, being the last two criteria applied for both, probes and samples. Additionally, sexual chromosomes, cross-reactive probes and probes with overlapping SNPs from dbSNP v.147[45] were discarded. Methylation beta values were normalized by means of functional normalization. After quality control, 574 samples and 768,067 probes were selected. The entire process was performed with minfi and meffil R packages[46,47].

### Statistical analysis

Statistical analyses were performed with R software environment 4.1.3. Heatmaps were plotted by means of pheatmap R package and other plots by means of ggplot2 R package, color scales and palettes were obtained from ggsci R package.

**Deconvolution of cell proportions.** Iterative hierarchical procedure implemented in EpiDISH R package[48] was used to estimate the main blood cell type proportions from methylome information with the robust partial correlation method[49]. Whole blood cell type reference panel includes: neutrophils, monocytes, B-lymphocytes, CD4 + T-Lymphocytes, CD8+ T-Lymphocytes and natural killer cells.

**Differential and interaction analysis.** Differential methylation analyses were performed by linear regression models, including age, sex and deconvoluted cell-proportions as covariates. Linear regression models including interaction terms between the groups of interest and deconvoluted cell proportions, were used to estimate the specific cell type(s) where the methylation changes occur, as proposed by Zheng et al.[50]. Methylation changes and interactions were considered significant at nominal $p$-values below 0.01 in discovery and replication datasets, and below a genome wide significant level of 5e-08 in the meta-analysis of both cohorts. Meta-analyses were performed with the restricted maximum likelihood (REML) method and fixed effects implemented in metafor R package[51].

**Enrichment, correlation and co-localization analysis.** DMCs (Differentially methylated CpGs) and/or genes that co-localized with them, based on the Illumina annotation (ilm10b4.hg19 R package), were analyzed. Functional pathway analysis was performed against Reactome Pathway Database[52] using ReactomePA R package[53] (genes covered by the 768,067 selected probes were set as background). CpG probe-oriented analysis was performed by means of the gsameth function from the missMethyl R package[54]. EWAS trait enrichments were tested within the EWAS Atlas database[23]. PRECISESADS methylomes[27] from seven SADs (SLE, systemic lupus erythematosus; RA, rheumatoid arthritis; pSjS, primary Sjögren's syndrome; SSc, systemic sclerosis; MCTD, mixed connective tissue disease; PAPS, primary anti-phospholipids syndrome and UCTD, undifferentiated connective

tissue disease) were used to compare with COVID-19 epigenetic changes. In order to compare both datasets, the PRECISESADS and the COVID-19, the methylation value of each probe was normalized by calculating the log2 fold-change with PRECISESADS healthy controls and PCR negative individuals, respectively. TFBS (transcription factor binding site) motif enrichment analysis was performed with HOMER software[55] using a size of 200 nucleotides and including as background the CpGs interrogated with the EPIC array.

**Molecular pathway activity analysis.** Single-cell RNA-Seq datasets were obtained from Schulte-Schrepping et al.[12] (BD Rhapsody system dataset, including neutrophils) and Ren et al.[11] (10x Genomics chromium dataset, not including neutrophils). Cells from both datasets were selected based on: mitochondrial read percentage < 5%, hemoglobin read percentage < 1%, number of reads > 500 and < 6000, and number of genes profiled between 200 and 2000. After the quality criteria filtering, almost all non-neutrophil cells were lost from Schulte-Schrepping et al. dataset. Thus, CD8+ T-lymphocytes and B-lymphocytes were analyzed from the Ren et al. dataset and neutrophils from the Schulte-Schrepping et al. Individuals were classified as early or late based on Schulte-Schrepping et al. definition (late, sampling > 11 days after first symptoms) and authors defined cell-type annotation was used to select two subsamples of 2500 cells for each cell-type (500 cells per severity group and onset category). Molecular pathway activity values were estimated by means of ssgsea algorithm implemented in escape R package[56]. HLA and Immunoglobulin genes were removed from the Reactome pathways before activity estimation.

**Genetic statistical analyses.** Overall genetic contribution to DNA methylation changes (heritability, h2) was estimated by means of two models: one based on variance decomposition analysis from a linear mixed-model[57] and the other one using the diagonalization trick[58]. The kinship matrix for the former model was calculated by means of popkin R package[59], while for the diagonalization trick estimation, gaston R package recommendations were followed[58]. Methylation quantitative trait loci (meQTLs) analyses were performed using the matrix-eQTL R package[60]. We applied a linear regression model that tests the additive effects of allele dosages for each genetic variant on the DNA methylation levels, while correcting for age, sex, the deconvoluted cell proportions and the first two genetic principal components. We restricted analysis to cis-meQTL mapping (maximum distance between CpG and SNPs of 1 Mb) and SNPs with minor allele frequencies (MAF) > 0.05. cis-meQTL analyses were performed independently on the different groups, using a FDR < 0.05 as significance threshold. Significant meQTLs were classified as common or specific QTLs based on whether the association nominal $p$-values were below 0.05 for all the groups or not. Then classifying non-common QTLs based on the groups that pass the threshold (QTL effects were took into consideration which might result in shared significant QTLs between groups but with opposite effects). MeQTLs enrichments were tested against SNP associated traits from the GWAS catalog database[29] expanded with COVID-19 Host Genetics Initiative results[7,8]. GWAS catalog traits were selected based on studies with a replication cohort and at least 50 SNPs below the genomic significant threshold ($p$-value < 5e-08). Traits annotation into meQTLs were performed based on linkage-disequilibrium blocks by means of PLINK1.9 software[61], applying blocks function[62] default parameters in a maximum window size of 1 MB.

**Reporting summary**
Further information on research design is available in the Nature Research Reporting Summary linked to this article.

## Data availability
Genotypes summary statistics can be accessed through COVID-19 Host Genetic Initiative web page (https://www.covid19hg.org/), included in

the project "Determining the Molecular Pathways and Genetic Predisposition of the Acute Inflammatory Process Caused by SARS-CoV-2 (SPGRX)". The genotype data generated (SPGRX cohort) in this study have been deposited in the European Genome-phenome Archive (EGA) database under accession code EGAS00001005304. The methylation data generated in this study have been deposited in the Gene Expression Omnibus (GEO) database under accession code GSE179325. The clinical data collected in this study are provided in Supplementary Data 2. The additional methylation data used in this study are available in the GEO database under accession code GSE167202. The scRNA-Seq data used in this study are available in the EGA database under accession code EGAS00001004571 and in GEO database under accession code GSE158055. Source data are provided with this paper.

## Code availability
No custom code or unpublished methods were used in the study. The scripts used in the generation of this manuscript are available upon request.

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

## Acknowledgements

This work has been supported through Consejería de Transformación Económica, Industria, Conocimiento y Universidades of the regional government of Andalucía cofounded by the European Union through European Regional Development Fund to MEAR (FEDER, CV20-10150), Consejo Superior de Investigaciones científicas (CSIC-COV19-016/202020E155) and Junta de Castilla y León (Proyectos COVID 07.04.467B04.74011.0 and Programa Estratégico Instituto de Biología y Genética Molecular, IBGM excellence programme references CLU-2029-02 and CCVC8485) to D.B., D.B. is also part of the CSIC's Global Health Platform (PTI Salud Global), Consejería de Salud y Familias of the regional government of Andalucía (PECOVID-0072-2020) to E.C.M. G.B. is supported by the Instituto de Salud Carlos III (ISCIII, Spanish Health Ministry) through the Sara Borrell subprogram (CD18/00153). The authors would like to particularly express their gratitude to the patients, nurses and many others who helped directly or indirectly in the consecution of this study.

## Author contributions

M.E.A.R. and G.B. designed the study; S.R.R., B.S., and D.B. recruited the patients, performed their clinical assessment, and obtained the samples; O.P.P. collected and quality controlled patient comorbidities from medical record data; C.A.D. performed genotyping and DNA methylation profiling; M.M.B. performed genotype data curation, imputation, and processing; G.B. performed DNA methylation data curation and processing; G.B. and E.C. performed data analyses; G.B., E.C., M.M.B., and M.E.A.R. discussed and interpreted the results; and G.B. and M.E.A.R. wrote the entire manuscript. All authors approved of the content of the manuscript.

## Competing interests

The authors declare no competing interests.
