## [Peer Review File · Nature Communications]

Whole-Blood DNA Methylation Analysis Reveals Respiratory Environmental Traits Involved in COVID-19 Severity Following SARS-CoV-2 InfectionREVIEWER COMMENTS

Reviewer #1 (Remarks to the Author):

Whole-Blood DNA Methylation Analysis Reveals Respiratory Environmental Traits Involved in COVID-19 Severity Following SARS-CoV-2 Infection

Barturen et al.

The authors performed an EWAS study combined with DNA genotyping in SARS-CoV-2 positive (n=473) and negative (n=101) cases. The analyses and interpretations of the data are problematic.

The most significant findings are those derived from the comparison between mild and severe COVID-19 cases. Considering that the comparisons with the control negative cases (mild vs. negative, severe vs. negative) do not provide major significant findings, they could be presented as supplementary information, even more considering that the identified DNA methylation changes are not specific for COVID-19, as they have been previously associated with autoimmune, allergy and asthma conditions, as well as respiratory-related environmental exposure or lifestyle habits (Fig. 2C). In this way, the mild vs. severe results could be further emphasized and clearly presented in the main manuscript.

Clinical data provided in Table 1 should be extended. In addition to age and gender, it is particularly important to revise and consider the comorbidities that have already been described as risk factors for severity in COVID-19, including diabetes, obesity, hypertension and chronic pulmonary and cardiovascular diseases. Patients could even be categorized using the Charlson Comorbidity Index (CCI), if is the case. If significant differences are detected between groups, those features should be considered as covariates in the differential methylation analysis, as the authors accurately performed for age and gender.

Regarding the functional enrichment analysis (Fig. 2A), in addition to the direction of the effects (Hypo- or hypermethylation), the genomic locations should be considered. As denoted by the authors, hypo- and hypermethylation events in CpGs located at 5'-regulatory regions are strongly associated to gene activation or silencing, respectively. However, the opposite scenario is frequently observed in CpGs located in gene body. Considering around 50-60% of the DMC are located around TSS and 5'UTRs in most of the comparisons (according Supp. Fig. 2B), pre-selection of those promoter-associated CpGs could further expand the biological meaning of the functional enrichment analysis. The same consideration could be applied for the identification of differentially methylated CpGs with significant interactions for the deconvoluted cell-type proportion (Fig. 2B).

Regarding the identified 4 DMC modules, could you hypothesize why the methylation values in the negative control cases are similar to the ones observed in severe cases in the M.He and M.Ho modules (in clear contrast to the methylation scores found in mild cases)? Considering the scores in negative cases, the description and use of the M.He module as the one that better differentiate severe and mild cases should be revised. In the same line, what about the respiratory environmental contribution to M.He considering the similar scores in severe and negative cases? Is the relationship between environmental exposure and the cytokine storm associated with the most critical outcomes of COVID-19 disease, suggested by the authors, supported by the epigenomic landscape found in negative cases (considering that the finding derives from the M.He module/Fig. 3C)? Maybe the statement emphasized in the Abstract section should be toned down, depending of the findings in negative controls. "We found an environmental trait-related signature that discriminates mild from severe cases, and regulates IL-6 expression via the transcription factor CEBP. The analyses suggest that an interaction between environmental contribution, genetics and epigenetics might be playing a role in triggering the cytokine storm described in the most severe cases".

Moreover, could you further describe in Methods section the negative cases included in the study? Are

they healthy individuals? Also provide the key information about the used WHO COVID-19 classification (and the reference) in Methods/Reference sections.

Further analyses, as performed for M.He and M-Ho should be performed in S.Ho and S.He modules, as they clearly discriminate severe vs. mild, but also severe and mild vs. negative cases; boosting the biological and clinical significance of the findings. Both, results from enrichment analyses shown in Fig. 3B, as well as the strikingly significant overrepresentation of interferon-stimulated response element (ISRE) motif in S.Ho module (Fig. 3D) encourage to further investigate S.Ho and S.He modules. Among the 4 modules described in Fig. 3, it is surprising that the S.He also reaches the highest enrichment score (Pearson correlation) in mild cases (Fig. 3E). It is apparently unexpected, if compared with Fig. 3A. In the other three modules (S.Ho, M.Ho and M.He), contrasting scores in mild and severe groups are found (as expected). For comparison, could you add the negative control group in Fig. 3E?

Inclusion of negative control cases in the severity stratification is confusing, particularly in Fig. 3A, as it seems to denote COVID-19 severity that is not the case. Consider to denote them as negative cases.

About Fig. 3B, "Potential therapeutics for SARS" as reactome significant pathway is a very interesting finding. To provide details about the gene-associated to the DMC annotated in that category could be useful for the clinical setting and shed lights for developing novel and/or improving existing therapies.

Rephrase the first paragraph in Introduction section, as COVID-19 is the disease (asymptomatic, mild or severe), and SARS (severe acute respiratory syndrome) the syndrome.

Revise the time of sampling in Methods section, as it is described that the samples were taken within a week after first symptoms, but early and late sampling are denoted in the manuscript (i.e. Supp. Fig. 4).

In addition to the cited references, the introduction/discussion should include recent important GWAS and EWAS studies in COVID-19 that have provided pivotal evidences about the genetic and epigenetic traits linked to disease susceptibility (Covid-19 GWAS Group Severe et al., NEJM 2020; Bastard et al., Science 2020; Zhang et al., Science 2020; Pairo-Castineira et al., Nature 2020; Castro-de-Moura et al., EBioMedicine 2021; Balnis et al., Clin Epigenetics 2021).

Reviewer #2 (Remarks to the Author):

Barturen et al. have performed an EWAS in peripheral blood from 473 and 101 SARS-CoV-2 lab positive and negative tested individuals, respectively. This was split into a discovery set (subset n = 47 negative, 269 mild, and 98 severe disease) and a replication set (subset n = 54 negative, 91 mild, and 15 severe disease).

The blood cell type deconvolution identified an increase in neutrophils with severe disease, as previously observed. As well, significant differences in age and gender were seen. Differential analysis was performed between the subsets and in a severity analysis by according the subsets' severity to a numerical scale. The authors identified in total 530 DMCs that replicated across the two sets. 43 DMCs were found in the severe versus negative comparison, 347 in the mild versus negative, 20 in the severe versus mild, and, finally, 257 in the severity analysis.

Overall, 24 DMCs, annotated into 17 different genes were shared between severe versus negative, mild versus negative, and with the severity analysis. These shared signals were identified to be involved in the activation of viral defense type I interferon inducible genes, the hyperactivation of B and T lymphocytes, as well as genes known to interact with ACE2. DMCs were also grouped by hierarchical clustering, which suggested a relationship between environmental factors and the severe covid related cytokine storm.

Within the remits of the available data, this is has been run as a methodologically sound EWAS.

However, I have some major points for the authors to consider below.

Major

1. In the Introduction the authors state that the DNA hypermethylation and hypomethylation in 5'-end regions of genes is directly related to the inactivation and activation of gene expression. However, this statement needs to be more nuanced, as whilst a DNA hypomethylated CpG-dense promoter is permissive for expression, this is not in itself sufficient to predict active expression [1].
2. Did the authors calculate the power of the discovery and replication datasets? – which are small by EWAS standards [2]. How might the power of the relative datasets impact the conclusions drawn and the confidence in their results?
3. The deconvolution analysis identified a significant increase in Neutrophils - as the authors state - previously identified haematologically to be associated with covid severity. However, could the authors potentially explore this result further by more fine-grained deconvolution of the neutrophil subgroups - as this signal is driven specifically by an increase in the immature neutrophil population [3].
4. The ageing-related signal in the DNA methylome is strong and ubiquitous. There are significant differences in age profile in the severe subgroup of the discovery set. The authors removed this influence statistically by including age as a covariate – but have any of their resulting DMCs previously been identified to change in the same direction with age?
5. A functional enrichment analysis was performed based on the Reactome pathway. Was this analysis explicitly corrected for the probe distribution bias of the DNA methylation EPIC array? [4]
6. Regarding the genetic effects on DNA methylation – the authors correctly removed confounded and SNP overlapping probes. However, the genetic analysis should also include further delineate those signals that are potentially confounded strongly by genetic/allelic effects - through the identification of DNA methylation trimodal clustering by methods such as GapHunter [5]. Also, were any unreliable highly variable probes recently identified by Sugden et al. included in the DMCs? [6].
7. The DMCs were grouped by hierarchical clustering, but did the authors attempt a direct regional analysis for DMRs? – as these regional changes can be enriched for functionality.

Minor

1. Strictly speaking the second dataset analysis should be referred to as 'Replication' not 'Validation' - as not performed by a differing methodology
2. Spelling "Crohn's" Fig 4 E
3. DNA Methylation array data not yet available at GEO

1. Schubeler, D., Function and information content of DNA methylation. *Nature*, 2015. 517(7534): p. 321-326.
2. Mansell, G., et al., Guidance for DNA methylation studies: statistical insights from the Illumina EPIC array. *BMC Genomics*, 2019. 20(1): p. 366.
3. Reusch, N., et al., Neutrophils in COVID-19. *Front Immunol*, 2021. 12: p. 652470.
4. Maksimovic, J., A. Oshlack, and B. Phipson, Gene set enrichment analysis for genome-wide DNA methylation data. *Genome Biol*, 2021. 22(1): p. 173.
5. Andrews, S.V., et al., "Gap hunting" to characterize clustered probe signals in Illumina methylation array data. *Epigenetics Chromatin*, 2016. 9: p. 56.
6. Sugden, K., et al., Patterns of Reliability: Assessing the Reproducibility and Integrity of DNA Methylation Measurement. *Patterns*, 2020.

Reviewer #1

1. *“The most significant findings are those derived from the comparison between mild and severe COVID-19 cases. Considering that the comparisons with the control negative cases (mild vs. negative, severe vs. negative) do not provide major significant findings, they could be presented as supplementary information, even more considering that the identified DNA methylation changes are not specific for COVID-19, as they have been previously associated with autoimmune, allergy and asthma conditions, as well as respiratory-related environmental exposure or lifestyle habits (Fig. 2C). In this way, the mild vs. severe results could be further emphasized and clearly presented in the main manuscript.”*

We would like to make some clarifications to this point.

In first place, the comparison of each COVID-19 severity group against the negative control is crucial in our analyses. Hypermethylation specificity in mild cases in the M.He module could only be observed when including the negative individuals in the analysis. This finding is highly relevant and we believe should be highlighted, as such specific hypermethylation might play a protective role upon SARS-CoV-2 infection.

On the other hand, we are concerned about reviewer's point that “the identified DNA methylation changes are not specific for COVID-19 as they have been previously associated with other conditions”. Shared molecular changes across conditions does not downplay the importance of the findings for a particular disease. It is, on the contrary, very important to understand the relationship between asthma and allergy conditions and the severity of Covid-19, something that has not been considered, and particularly environmental pollutants. Moreover, enrichment in other traits only shows that some of the changes were also previously found to be regulating similar molecular pathways that are important during epigenetic regulation.

2. *“Clinical data provided in Table 1 should be extended. In addition to age and gender, it is particularly important to revise and consider the comorbidities that have already been described as risk factors for severity in COVID-19, including diabetes, obesity, hypertension and chronic pulmonary and cardiovascular diseases. Patients could even be categorized using the Charlson Comorbidity Index (CCI), if is the case. If significant differences are detected between groups, those features should be considered as covariates in the differential methylation analysis, as the authors accurately performed for age and gender.”*

We have made a big effort to reunite information from clinical records on fourteen comorbidities including: non-hematological cancer, asthma, chronic respiratory disease, chronic liver disease, chronic heart disease, hypertension, neurological disorder, stroke autoimmune disease, ever-smoker, current-smoker, hematological cancer history, asplenia and organ transplant were collected from the clinical records (the last three comorbidities were present in less than 5 cases in both cohorts and were not tested).

Only four comorbidities (asthma, chronic heart disease, hypertension and current-smoker) showed significant differences (p -value < 0.05) at least between two groups (mild, severe or negative COVID19 patients), either in discovery or replication cohort.

In order to study the extent of the effect of these comorbidities in our results, they were added to the regression models as covariates. Our analyses show that their inclusion did

not modify the results obtained in the original models. All corrected-DMCs remain significant at p -values on the meta-analysis below $5e-06$. In addition, the statistics from discovery, replication and the meta-analysis did not change, showing a high correlation of almost 1. We can therefore conclude that the comorbidities do not bias the methylation differences observed between groups in the original models.

The results from these corrected models have now been included in the manuscript in the results section: *“The influence of comorbidities on the results was tested by adding all comorbidity categories with a Fisher’s exact test p -value < 0.05 (between at least two groups either in the discovery or the replication cohort) in the linear models. These were asthma, chronic heart disease, hypertension and current smokers out of 14 tested. All DMCs remained significant at a p -value below $5e-06$ in the meta-analysis. The statistics for both discovery and replication models as well as for the meta-analysis showed a high correlation with an R -squared correlation ~ 1 and a p -value below $2.2e-16$ (see Supplementary Figures 1E, 1F and 1G).”* (Lines 100-106). The Fisher exact test results are included in the source data file, the corrected models’ statistics in the Supplementary File 1. Three additional figures showing the statistic correlations of the model were added as Figures 1E, 1F and 1G.

3. *“Regarding the functional enrichment analysis (Fig. 2A), in addition to the direction of the effects (Hypo- or hypermethylation), the genomic locations should be considered. As denoted by the authors, hypo- and hypermethylation events in CpGs located at 5’-regulatory regions are strongly associated to gene activation or silencing, respectively. However, the opposite scenario is frequently observed in CpGs located in gene body. Considering around 50-60% of the DMC are located around TSS and 5’UTRs in most of the comparisons (according Supp. Fig. 2B), pre-selection of those promoter-associated CpGs could further expand the biological meaning of the functional enrichment analysis.”*

Thanks for this interesting comment. We completely agree that the CpG location can impact how DNA methylation is related to gene expression. In the case of CpGs included in EPIC arrays, the intergenic probes have not been assigned to any gene, thus they do not contribute to the enrichment analysis. Then, the enrichment analysis is highly represented by CpGs at 5’-end regulatory regions. We are also aware that DNA methylation changes at 5’ regulatory regions do not always necessarily affect gene expression, thus sub-selecting 5’-end DMCs will not solve the biological interpretation of the results. Therefore, we considered that the best approach was to validate the DNA methylation effects into the enriched pathways by means of scRNA-Seq coming from previous publications (Supplementary Figure 4), and include this analysis in the manuscript.

4. *“The same consideration could be applied for the identification of differentially methylated CpGs with significant interactions for the deconvoluted cell-type proportion (Fig. 2B).”*

The deconvolution analysis was performed in order to investigate the cell-type contribution on the DNA methylation changes in whole-blood, and in this way focus the scRNA-Seq validation of the enriched pathways into the most affected cell types (Supplementary Figure 4). Considering this comment, the previous one, and the comments in point 14, we realize that the results from the scRNA-Seq analysis are not properly allocated in the flow of the manuscript, as they appear to generate confusion and make it hard to relate the analysis directly with the validation of the pathway regulation. Thus, we have moved these results from *“Respiratory environmental related epigenetic changes differentiate severe and mild COVID-19 patients and mild COVID-19 cases from systemic autoimmune disorders”* section to the previous one, *“COVID-19 disease DNA methylation*

changes in neutrophils, B-lymphocytes and CD8+ T-lymphocytes regulate functional pathways related with autoimmune and viral defenses”, where the related cell-types and pathways are found. Additionally, mention of the CpG modules has been removed from this section in order to adapt it to the logical flow in this new allocation.

5. *“Regarding the identified 4 DMC modules, could you hypothesize why the methylation values in the negative control cases are similar to the ones observed in severe cases in the M.He and M.Ho modules (in clear contrast to the methylation scores found in mild cases)? Considering the scores in negative cases, the description and use of the M.He module as the one that better differentiate severe and mild cases should be revised. In the same line, what about the respiratory environmental contribution to M.He considering the similar scores in severe and negative cases? Is the relationship between environmental exposure and the cytokine storm associated with the most critical outcomes of COVID-19 disease, suggested by the authors, supported by the epigenomic landscape found in negative cases (considering that the finding derives from the M.He module/ Fig. 3C)? Maybe the statement emphasized in the Abstract section should be toned down, depending of the findings in negative controls. “We found an environmental trait-related signature that discriminates mild from severe cases, and regulates IL-6 expression via the transcription factor CEBP. The analyses suggest that an interaction between environmental contribution, genetics and epigenetics might be playing a role in triggering the cytokine storm described in the most severe cases”.”*

Considering the results coming from the reliability analyses proposed by reviewer #2, and the lack of consistency of the DNA methylation patterns shown for negative controls in the M.Ho module (Fig. 3A), we removed this module from the functional analyses and conclusions of the manuscript. This module seems to be unstable, based on the additional analyses performed during the revision.

Regarding the differences found in module M.He, we hypothesized that specific hypermethylation of these CpGs might be playing a protective role during SARS-CoV-2 infection, as these changes are located in TF binding sites related with the positive regulation of cytokines that are, themselves, related to the most severe outcomes of the disease. Apart from CEBP, already mentioned in the manuscript, we also found that PU.1, Irf1, NKx6.1 and CREB5 might be also regulating the expression of IL-6 and other cytokines such as IL-1 α , IL-1 β and IL-12. Interestingly, IL-1 α has been proposed as an early marker of poor prognosis. Therefore, the hypermethylation in these regions with TF binding sites might represent a general protective process (as a large number of individuals do not develop a severe phenotype) on individuals not exposed to certain environmental conditions, comorbidities and/or genetic variants upon SARS-CoV-2 infection, while severe cases are not capable (by mechanisms out of the scope of this manuscript) to block or dysregulate these TFs interacting with DNA.

In order to clarify this conclusion, several sentences have been modified in the manuscript:

- *“The M.He module is hypermethylated in mild COVID-19 cases as compared with severe cases and negative controls, suggesting that differential respiratory environmental exposures might play a protective role against severe COVID-19 progression, upon SARS-CoV-2 infection.”* (lines 190-192)
- Fourth section title has been rephrased: *“Specific hypermethylation in mild cases shows a minor genetic contribution, while meQTLs are enriched in SNPs associated with environmental traits”*
- The statement in line 223 about the ability of M.He to discriminate mild/severe cases was removed as it was not tested in the manuscript, and substituted by *“As the specific mild hypermethylated changes...”*

- The TFBS information for module M.He has been expanded in the discussion and the conclusion rephrased as: *“Further analyses on this module revealed an enrichment in transcription factor binding sites (CEBP, PU.1, ISL1 and CREB), which are known to positively regulate the levels of cytokines related to COVID-19 severity as IL-6, IL-1 α , IL-1 β , IL-12 and other pro-inflammatory cytokines containing a cAMP-responsive elements (Figure 3D). Interestingly, IL-1 α has been proposed as an early marker of poor prognosis. The CEBP transcription factor has an important role regulating IL-6 and IL-1 β expression, whose elevated levels have been associated with severe complications of COVID-19 disease. The hypermethylation on M.He CpGs suggests a reduced binding activity of these transcription factors in mild cases compared to the severe cases and the negative controls, in a module where DMCs are enriched in respiratory environmental traits. Altogether, our results suggest the existence of a relationship between environmental exposure and the protection against cytokine storm associated with the most critical outcomes of COVID-19 disease.”* (lines 325-336)

6. “Moreover, could you further describe in Methods section the negative cases included in the study? Are they healthy individuals?”

A description of the negative cases has been included: *“Negative PCR individuals did not show obvious evidence of infection at sampling and none of them were admitted to the hospital.”* (lines 367-368)

7. *“Also provide the key information about the used WHO COVID-19 classification (and the reference) in Methods/Reference sections.”*

A mistake in the WHO scale has been corrected, a summary of the clinical ordinal scale has been added as Supplementary Table, 1 and the reference has been included. The sentence added in the methods section reads as follows: *“Individuals were classified based on the WHO clinical ordinal scale (Supplementary Table 1): PCR negative individuals (uninfected, 0 scale), mild PCR positive individuals (ambulatory or hospitalized with mild symptoms, 1-4 scales) and severe PCR positive individuals (hospitalized with severe symptoms or died, 5-8 scales).”* (lines 375-380)

8. *“Further analyses, as performed for M.He and M-Ho should be performed in S.Ho and S.He modules, as they clearly discriminate severe vs. mild, but also severe and mild vs. negative cases; boosting the biological and clinical significance of the findings Both, results from enrichment analyses shown in Fig. 3B, as well as the strikingly significant overrepresentation of interferon-stimulated response element (ISRE) motif in S.Ho module (Fig. 3D) encourage to further investigate S.Ho and S.He modules.”*

We believe there might be a misunderstanding. All the analyses were performed for all the modules (except for M.Ho which was removed as mention above due to bad reliability results). We want to add that the enrichment of interferon regulator binding sites was expected on a module enriched on interferon related genes, therefore, in a sense, this result serves as a positive control as it shows that the TFBS enrichment is working properly and provides good reliability to unexpected results, such as the ones coming from the M.He.

9. *“Among the 4 modules described in Fig. 3, it is surprising that the S.He also reaches the highest enrichment score (Pearson correlation) in mild cases (Fig. 3E). It is apparently unexpected, if compared with Fig. 3A. In the other three modules (S.Ho, M.Ho and M.He), contrasting scores in mild and severe groups are found (as expected).”*

The correlations from Figure 3E do not directly depend on the methylation values themselves but on the change when compared with controls, because we are comparing normalized methylation values using controls as reference (log₂FC). Then, the correlations will depend on the relationship between severe or mild against negative individuals and autoimmune disorders against their controls. Thus, the intensity of the change in the S.He module is similar to that of autoimmune disorders for both severe and mild, and the same happens for S.Ho in interferon related SADs. The difference in S.Ho for RA and SSc comes due to an additional signature (B/T hyperactivation) grouped into the S.Ho module as investigated and described in the manuscript. So, yes, there are signatures that correlate with SADs in both mild and severe cases, and others than do not. And potentially, the environmental M.He-related signature and the hyperactivation of B/T lymphocytes might be playing an important role during COVID-19 severity, as the interferon and phagocytosis pathways are also dysregulated in mild cases.

10. *“For comparison, could you add the negative control group in Fig. 3E?”*

The information on the negative control group is already included in this Figure. Comparing molecular information directly across experiments is challenging, because differences might arise due to batch effects and not by true differences. To overcome this problem, and still be able to compare results from COVID-19 and from other autoimmune conditions, we took the approach of exploring the differences between each group and the negative controls, and assess their similarities based on log₂FC correlations.

To make sure that our approach is clear to readers, we have extended the explanation in methods section: *“In order to compare both datasets, the PRECISESADS and the COVID-19, the methylation value of each probe was normalized by calculating the log₂ fold-change with healthy controls and PCR negative individuals.”* (line 435-437)

11. *“Inclusion of negative control cases in the severity stratification is confusing, particularly in Fig. 3A, as it seems to denote COVID-19 severity that is not the case. Consider to denote them as negative cases.”*

We are grateful for this comment. The figure has been modified accordingly.

12. *“About Fig. 3B, “Potential therapeutics for SARS” as reactome significant pathway is a very interesting finding. To provide details about the gene-associated to the DMC annotated in that category could be useful for the clinical setting and shed lights for developing novel and/or improving existing therapies.”*

Thanks for this suggestion. The genes tagged in each enriched pathway are now added in the Source Data file.

13. *“Rephrase the first paragraph in Introduction section, as COVID-19 is the disease (asymptomatic, mild or severe), and SARS (severe acute respiratory syndrome) the syndrome.”*

The sentence has been rephrased as: *“...while in others, the virus causes a disease called COVID-19 that primarily affects the lungs...”* (lines 41-42)

14. *“Revise the time of sampling in Methods section, as it is described that the samples were taken within a week after first symptoms, but early and late sampling are denoted in the manuscript (i.e. Supp. Fig. 4).”*

This analysis was performed in order to validate the effect of DNA methylation changes in the enriched pathways per cell-type (as already mentioned in points 3 and 4). scRNA-Seq data does not come from our samples. The datasets were downloaded from previous publications cited in the manuscript, and these samples were divided into early/late samplings (as described in the methods section, line 447) in order to compare them with our cohorts' early samplings.

This is described as follows in the results section: *“In order to validate the activation or inactivation of the enriched pathways as revealed by the DNA methylation changes, Reactome pathways' activity was estimated based on single-cell RNA-Seq information from publicly available analyses.”* (lines 139-142) *“In general, molecular pathway activities follow the DNA methylation changes at early sampling time points, which correspond with our recruited cohorts.”* (lines 143-145)

... and in the methods section: *“Single-cell RNA-Seq datasets were obtained from Schulte-Schrepping et al. (BD Rhapsody system dataset, including neutrophils) and Ren et al. (10x Genomics chromium dataset, not including neutrophils).”* (lines 441-443).

The results have been moved in order to put them in the right context in the manuscript, as already mentioned in point 4.

15. *“In addition to the cited references, the introduction/discussion should include recent important GWAS and EWAS studies in COVID-19 that have provided pivotal evidences about the genetic and epigenetic traits linked to disease susceptibility (Covid-19 GWAS Group Severe et al., NEJM 2020; Bastard et al., Science 2020; Zhang et al., Science 2020; Pairo-Castineira et al., Nature 2020; Castro-de-Moura et al., EBioMedicine 2021; Balnis et al., Clin Epigenetics 2021).”*

We have included all new suggested references and updated some of them that were previously added as pre-prints.

Reviewer #2

1. *“In the Introduction the authors state that the DNA hypermethylation and hypomethylation in 5’-end regions of genes is directly related to the inactivation and activation of gene expression. However, this statement needs to be more nuanced, as whilst a DNA hypomethylated CpG-dense promoter is permissive for expression, this is not in itself sufficient to predict active expression.”*

We agree with the reviewer. The sentence has been modified accordingly: *“hypermethylation and hypomethylation in 5’-end regions of the genes are mostly related to the inactivation and activation of gene expression, respectively”* (line 114)

2. *“Did the authors calculate the power of the discovery and replication datasets? – which are small by EWAS standards. How might the power of the relative datasets impact the conclusions drawn and the confidence in their results?”*

We agree that having a higher sample size would increase the power of our study, and impact our findings. However, we believe that we are close to EWAS standards. Considering our initial power analyses, we estimated the need of 112 cases and controls in order to reach an 80% EWAS power rate for a 10% mean methylation difference (PMID: 25972603). In our case, the smallest sample size is found in the severe/negative cross-sectional analysis, with 113 severe cases and 101 PCR negative individuals (considering both cohorts). Obviously, we would like to have a larger sample, but this was the number of patients that we were able to obtain with good DNA quality and within the first week after the first symptoms. Nevertheless, we did consider this limitation from the start of the study, and this is why we decided to meta-analyze both cohorts instead of merging them in a combined EWAS (which doubled the number of significant DMCs at the same threshold). In this way, we intended to gain reliability of the results. Additionally, we considered that the results coming from the functional analyses performed on the DMCs present a biological framework in the line of what is known about the disease reinforcing their reliability.

Nevertheless, we consider that sample size is a limitation of our study, and we have discussed this in the manuscript. We also tested the reliability of the identified DMCs based on the Sugden et al. metric (as proposed by the reviewer). Additionally, the log₂FC between groups was further replicated in an additional dataset from a paper published after our submission (Supplementary Figure 5). We have added to the discussion: *“In addition to the COVID-19 EWAS results, considering that our cohort is barely below EWAS size standards, and in order to filter out potential false positive results, DMCs were grouped by hierarchical clustering and filtered by cohorts’ similarity, reliability and replication with an external cohort (Figure 3 and Supplementary Figure 5).”* (lines 310-313)

We believe that these new reliability and replication analysis make our results robust.

3. *“The deconvolution analysis identified a significant increase in Neutrophils - as the authors state - previously identified haematologically to be associated with covid severity. However, could the authors potentially explore this result further by more fine-grained deconvolution of the neutrophil subgroups - as this signal is driven specifically by an increase in the immature neutrophil population.”*

We agree with the reviewer that the neutrophil population expansion associated with COVID-19 severity is likely related with the immature population (low density granulocytes), as previously described and properly cited in the manuscript.

Unfortunately, there is no methylome reference panel available for the deconvolution of immature neutrophils, and thus deconvolution on neutrophil subtypes should be performed by means of unsupervised deconvolution.

Following reviewer's recommendation, we used the DecompPipeline deconvolution protocol (Scherer et al. Nature Protocols, 2020). Briefly, the DNA methylation matrix was adjusted by independent components associated with age, gender or batch (ICA), the highest 2000 variable CpGs were loaded into MeDeCOM to explore the optimal number of components (2:15) and the lambda parameter (0.01, 1e-03, 1e-04 and 1e-05) based on cross-validation error. Then, LMC proportions for 7 components were estimated at a lambda parameter of 1e-03. The analysis was run on all samples together and then these were divided to test the replicability of the proportions between discovery and replication cohorts. First, LMC proportions were correlated with the supervised proportions obtained in the manuscript in order to determine the relationship between the LMCs and the supervised cellular proportions. As the reviewer can see in the plot below (A), we can distinguish two LMCs correlating neutrophil proportions (3 and 7), which might represent two different neutrophil populations. Unfortunately, we were not able to replicate the LMC proportions between our cohorts, thus we did not go ahead with the functional characterization and this result has not been included in the manuscript

4. "The ageing-related signal in the DNA methylome is strong and ubiquitous. There are significant differences in age profile in the severe subgroup of the discovery set. The authors removed this influence statistically by including age as a covariate – but have any of their resulting DMCs previously been identified to change in the same direction with age?"

When we tested DMC enrichments on EWAS atlas traits we found that 15 DMCs (out of the significant 540) overlapped with the aging trait (which covers 31 studies and 33086 probes). However, as shown in figures 2C and 3C the DMCs did not show a significant enrichment. Actually, even if we take a look into the non-significant results, there would be a depletion instead of an enrichment for our DMCs set, as the fractions are 15/540 (~0.03) vs 33086/~800.000 (~0.04).

5. "A functional enrichment analysis was performed based on the Reactome pathway. Was this analysis explicitly corrected for the probe distribution bias of the DNA methylation EPIC array?"

Yes, the genes covered by the quality-controlled EPIC probes were set as background. This information has been added to the methods section: “(*genes covered by the 768,067 interrogated probes were set as background*)” (lines 429-430)

6. “Regarding the genetic effects on DNA methylation – the authors correctly removed confounded and SNP overlapping probes. However, the genetic analysis should also include further delineate those signals that are potentially confounded strongly by genetic/allelic effects - through the identification of DNA methylation trimodal clustering by methods such as GapHunter.”

As proposed by the reviewer, GapHunter was run on our dataset. 15808 were found to present gap signals based on the default thresholds, and out of them only one (cg17868815) was found to be differentially methylated in our analysis. This probe was found to be differentially methylated on the cross-sectional mild-negative model and was not grouped into the CpG modules identified in the manuscript. Thus, gap signals have no impact on the results.

7. “Also, were any unreliable highly variable probes recently identified by Sugden et al. included in the DMCs?”

Thanks for pointing out this interesting paper. Considering that the *Sugden et al.* reliability values were calculated on blood samples, we estimated the reliability of our results by subsetting the values into the significant DMCs and the CpG modules found during the analysis. Then, the distributions were compared with the reliability values distribution of the CpGs included in the *Sugden et al.* paper by means of a Kolmogorov-Smirnov test. We found that the DMCs showed a significantly higher amount of reliable CpGs than the 450k probes, and, on the other hand, the clustering procedure improved the reliability of the results for clusters S.Ho and M.He. However, the M.Ho cluster showed a very low reliability, which might be expected, as the methylation pattern in the negative controls could not be replicated between discovery and replication cohorts (Figure 3A). Thus, a new plot has been added as Figure 3B, including the density plots of the reliability value distributions. Additionally, due to its low reliability, module M.Ho has been removed from subsequent analyses and individual reliability values have been added in the supplementary DMC information.

8. “The DMCs were grouped by hierarchical clustering, but did the authors attempt a direct regional analysis for DMRs? – as these regional changes can be enriched for functionality.”

Regional analysis has not been performed, as we believe that DMR analysis on methylation array data has some important limitations. First, because DMR analysis assumes that all the probes in the arrays are comparable across regions, but this is not the case as type I and II probes have a different chemistry. Second, not all the regions across the genome are equally covered by probes, what would cause and undesired bias on the results.

9. “Strictly speaking the second dataset analysis should be referred to as ‘Replication’ not ‘Validation’ - as not performed by a differing methodology”

Agree, we have corrected this wording across the entire manuscript.

10. “Spelling “Crohn’s” Fig 4 E”

Thanks, this typo error has been corrected.

11. *"DNA Methylation array data not yet available at GEO"*

The dataset is available now (GSE179325), but restricted until the manuscript is published. The id has been added to the data availability section.

REVIEWER COMMENTS

Reviewer #1 (Remarks to the Author):

Whole-Blood DNA Methylation Analysis Reveals Respiratory Environmental Traits Involved in COVID-19 Severity Following SARS-CoV-2 Infection

Barturen et al.

The revised manuscript has successfully addressed my comments and suggestions.

Only one point should be further discussed, in line with previous publications (Rishi et al, PNAS, 2010; Yang et al., Nucleic Acids Res. 2019; Sayeed et al., Biochim Biophys Acta. 2015), mainly considering that the authors' hypothesis has not been experimentally validated (i.e. BisChIP-seq).

The authors state "The CEBP transcription factor has an important role regulating IL-6 and IL-1 β expression, whose elevated levels have been associated with severe complications of COVID-19 disease. The hypermethylation on M.He CpGs suggests a reduced binding activity of these transcription factors in mild cases compared to the severe cases and the negative controls, in a module where DMCs are enriched in respiratory environmental traits". Whereas, some previous works have shown that CpG methylation of the binding sites enhances the DNA binding of the C/EBP transcription factors. Please discuss.

Reviewer #2 (Remarks to the Author):

>>>Reviewer #2 Replies

1."In the Introduction the authors state that the DNA hypermethylation and hypomethylation in 5'-end regions of genes is directly related to the inactivation and activation of gene expression. However, this statement needs to be more nuanced, as whilst a DNA hypomethylated CpG-dense promoter is permissive for expression, this is not in itself sufficient to predict active expression."

We agree with the reviewer. The sentence has been modified accordingly:

"hypermethylation and hypomethylation in 5'-end regions of the genes are mostly related to the inactivation and activation of gene expression, respectively" (line 114)

>>>Ok

2."Did the authors calculate the power of the discovery and replication datasets?—which are small by EWAS standards. How might the power of the relative datasets impact the conclusions drawn and the confidence in their results?"

We agree that having a higher sample size would increase the power of our study, and impact our findings. However, we believe that we are close to EWAS standards. Considering our initial power analyses, we estimated the need of 112 cases and controls in order to reach an 80% EWAS power rate for a 10% mean methylation difference (PMID: 25972603). In our case, the smallest sample size is found in the severe/negative cross-sectional analysis, with 113 severe cases and 101 PCR negative individuals (considering both cohorts). Obviously, we would like to have a larger sample, but this was the number of patients that we were able to obtain with good DNA quality and within the first week after the first symptoms. Nevertheless, we did consider this limitation from the start of the study, and this is why we decided to meta-analyze both cohorts instead of merging them in a combined EWAS (which doubled the number of significant DMCs at the same threshold). In this way, we intended to gain reliability of the results. Additionally, we considered that the results coming from the functional analyses performed on the DMCs present a biological framework in the line of what is known about the disease reinforcing their reliability.

Nevertheless, we consider that sample size is a limitation of our study, and we have discussed this in

the manuscript. We also tested the reliability of the identified DMCs based on the Sugden et al. metric (as proposed by the reviewer). Additionally, the log₂FC between groups was further replicated in an additional dataset from a paper published after our submission (Supplementary Figure 5). We have added to the discussion: "In addition to the COVID-19 EWAS results, considering that our cohort is barely below EWAS size standards, and in order to filter out potential false positive results, DMCs were grouped by hierarchical clustering and filtered by cohorts' similarity, reliability and replication with an external cohort (Figure 3 and Supplementary Figure 5)." (lines 310-313)

We believe that these new reliability and replication analysis make our results robust.

>>> The paper they have cited is for monozygotic power calculation so would appear not to be appropriate for the population analysis here? – authors should incorporate the more recent Mansell et al. (2019) BMC Genomics assessment of DNA methylation array power.

3. "The deconvolution analysis identified a significant increase in Neutrophils - as the authors state - previously identified haematologically to be associated with covid severity. However, could the authors potentially explore this result further by more fine-grained deconvolution of the neutrophil subgroups - as this signal is driven specifically by an increase in the immature neutrophil population."

We agree with the reviewer that the neutrophil population expansion associated with COVID-19 severity is likely related with the immature population (low density granulocytes), as previously described and properly cited in the manuscript.

Unfortunately, there is no methylome reference panel available for the deconvolution of immature neutrophils, and thus deconvolution on neutrophil subtypes should be performed by means of unsupervised deconvolution.

Following reviewer's recommendation, we used the DecompPipeline deconvolution protocol (Scherer et al. Nature Protocols, 2020). Briefly, the DNA methylation matrix was adjusted by independent components associated with age, gender or batch (ICA), the highest 2000 variable CpGs were loaded into MeDeCOM to explore the optimal number of components (2:15) and the lambda parameter (0.01, 1e-03, 1e-04 and 1e-05) based on cross-validation error. Then, LMC proportions for 7 components were estimated at a lambda parameter of 1e-03. The analysis was run on all samples together and then these were divided to test the replicability of the proportions between discovery and replication cohorts. First, LMC proportions were correlated with the supervised proportions obtained in the manuscript in order to determine the relationship between the LMCs and the supervised cellular proportions. As the reviewer can see in the plot below (A), we can distinguish two LMCs correlating neutrophil proportions (3 and 7), which might represent two different neutrophil populations. Unfortunately, we were not able to replicate the LMC proportions between our cohorts, thus we did not go ahead with the functional characterization and this result has not been included in the manuscript

>>> Good that the authors attempted to explore this further

4. "The ageing-related signal in the DNA methylome is strong and ubiquitous. There are significant differences in age profile in the severe subgroup of the discovery set. The authors removed this influence statistically by including age as a covariate – but have any of their resulting DMCs previously been identified to change in the same direction with age?"

When we tested DMC enrichments on EWAS atlas traits we found that 15 DMCs (out of the significant 540) overlapped with the aging trait (which covers 31 studies and 33086 probes). However, as shown in figures 2C and 3C the DMCs did not show a significant enrichment. Actually, even if we take a look into the non-significant results, there would be a depletion instead of an enrichment for our DMCs set, as the fractions are 15/540 (~0.03) vs 33086/~800.000 (~0.04).

>>> Excellent to investigate this in more detail

5. A functional enrichment analysis was performed based on the Reactome pathway. Was this analysis

explicitly corrected for the probe distribution bias of the DNA methylation EPIC array?"

Yes, the genes covered by the quality-controlled EPIC probes were set as background. This information has been added to the methods section: "(genes covered by the 768,067 interrogated probes were set as background)" (lines 429-430)

>>> Unfortunately, I think the authors have misinterpreted this – as just using the genes covered by the probes as background does not take into consideration that some genes have disproportionately more probes. Please refer to Maksimovic et al. (2021) Genome Biology for non-biased analysis.

6. "Regarding the genetic effects on DNA methylation – the authors correctly removed confounded and SNP overlapping probes. However, the genetic analysis should also include further delineate those signals that are potentially confounded strongly by genetic/allelic effects - through the identification of DNA methylation trimodal clustering by methods such as GapHunter."

As proposed by the reviewer, GapHunter was run on our dataset. 15808 were found to present gap signals based on the default thresholds, and out of them only one (cg17868815) was found to be differentially methylated in our analysis. This probe was found to be differentially methylated on the cross-sectional mild-negative model and was not grouped into the CpG modules identified in the manuscript. Thus, gap signals have no impact on the results.

>>> Excellent to thoroughly evaluate the genetic influence on results

7. "Also, were any unreliable highly variable probes recently identified by Sugden et al. included in the DMCs?"

Thanks for pointing out this interesting paper. Considering that the Sugden et al. reliability values were calculated on blood samples, we estimated the reliability of our results by subsetting the values into the significant DMCs and the CpG modules found during the analysis. Then, the distributions were compared with the reliability values distribution of the CpGs included in the Sugden et al. paper by means of a Kolmogorov-Smirnov test. We found that the DMCs showed a significantly higher amount of reliable CpGs than the 450k probes, and, on the other hand, the clustering procedure improved the reliability of the results for clusters S.Ho and M.He. However, the M.Ho cluster showed a very low reliability, which might be expected, as the methylation pattern in the negative controls could not be replicated between discovery and replication cohorts (Figure 3A). Thus, a new plot has been added as Figure 3B, including the density plots of the reliability value distributions. Additionally, due to its low reliability, module M.Ho has been removed from subsequent analyses and individual reliability values have been added in the supplementary DMC information.

>>> Excellent to include this evaluation

8. "The DMCs were grouped by hierarchical clustering, but did the authors attempt a direct regional analysis for DMRs? – as these regional changes can be enriched for functionality."

Regional analysis has not been performed, as we believe that DMR analysis on methylation array data has some important limitations. First, because DMR analysis assumes that all the probes in the arrays are comparable across regions, but this is not the case as type I and II probes have a different chemistry. Second, not all the regions across the genome are equally covered by probes, what would cause and undesired bias on the results.

>>> Ok – although would have been interesting to explore, despite the known imperfections, to see if any biologically interesting results arose as DMRs can be more easily interpretable. Furthermore, methods exist to correct the probe type variation (BMIQ, Teschendorff et al.)

9. "Strictly speaking the second dataset analysis should be referred to as 'Replication' not 'Validation' - as not performed by a differing methodology"

Agree, we have corrected this wording across the entire manuscript.

>>> Great

10. "Spelling "Crohn's" Fig 4 E"

Thanks, this typo error has been corrected.

>>> Great

11. "DNA Methylation array data not yet available at GEO"

The dataset is available now (GSE179325), but restricted until the manuscript is published. The id has been added to the data availability section.

>>> Great

Before answering the remaining comments on the manuscript, we want to thank for the thorough revision made by both reviewers, and the interesting and important points raised by them, which we really feel that have importantly contributed to improve the scientific quality of the manuscript and the impact of its results.

REVIEWER COMMENTS

Reviewer #1

1. Only one point should be further discussed, in line with previous publications (Rishi et al, PNAS, 2010; Yang et al., Nucleic Acids Res. 2019; Sayeed et al., Biochim Biophys Acta. 2015), mainly considering that the authors' hypothesis has not been experimentally validated (i.e. BisChIP-seq).

The authors state "The CEBP transcription factor has an important role regulating IL-6 and IL-1 β expression, whose elevated levels have been associated with severe complications of COVID-19 disease. The hypermethylation on M.He CpGs suggests a reduced binding activity of these transcription factors in mild cases compared to the severe cases and the negative controls, in a module where DMCs are enriched in respiratory environmental traits". Whereas, some previous works have shown that CpG methylation of the binding sites enhances the DNA binding of the C/EBP transcription factors. Please discuss.

Thanks for raising this interesting point. Summarizing the citations mentioned by the reviewer, it seems that the CEBPa binding affinity for methylated sites might be context dependent. In Rishi et al., 85% of the promoters bound by CEBPa contained the consensus binding motif and are unmethylated (the consensus motif is the one enriched in our homer analysis: TTGCGCAA, not the CRE sequence, mentioned in the PNAS title: TGACGTCA, you can check it in the source data file). In the remaining 15% of methylated promoters, upon demethylation, CEBPa activity is decreased (but their motif is the CRE sequence not the consensus one, check Figure 2E of the PNAS manuscript). On the other hand, the other two articles refer to CEBPb, whose interaction is enhanced by 5mC and blocked by 5hmC (not by unmethylated Cs). Unfortunately, the bisulfite conversion used in methylation assay cannot distinguish between 5hmC and 5mC. Thus, potentially the CpGs of the M.He located in the CEBP consensus binding sequence, are detecting its oxidative form and not the 5mC, thus blocking in mild cases, the interaction of CEBPb. Another article found that IL-6 treatment mediated the demethylation of CEBPb binding sites followed by an elevated interaction of this TF (Wang et al., Hypertension 2013). This suggests that on an IL-6 stimulation context the CEBPb binds to unmethylated domains. It is also known that CEBPb plays an important role in the positive regulation of IL-6 levels (Kuilman et al., Cell 2008), thus, creating a positive feedback loop towards the overexpression of IL-6. All this makes it plausible that increased 5hmC (detected as 5mC in our analysis) might be blocking the interaction for CEBPb, while the unmethylated domains allow CEBPb interact with the DNA. In addition, other TFs were also enriched in the M.He module, which might show a similar regulation or blockade by 5mC.

Taking into consideration that we cannot affirm with experimental support that the binding activity is reduced in all the M.He sites, we agree with the reviewer that the hypothesis proposed should be tuned down to: *"The hypermethylation on M.He CpGs suggests a differential binding activity..."* instead of *"The hypermethylation on M.He CpGs suggests a reduced binding activity..."* (lines 334-335).

Reviewer #2

2. The paper they have cited is for monozygotic power calculation so would appear not to be appropriate for the population analysis here? – authors should incorporate the more recent Mansell et al. (2019) BMC Genomics assessment of DNA methylation array power.

Actually, both MZ twins and case-control powers were calculated in the reference cited raising a similar results (Figure 3 for case-control and Figure 4 for MZ twins) and the numbers given in the previous revision were for the case-control (Table 2 of the same manuscript). Nevertheless, we ran the more recent tool proposed by the reviewer, showing that the power for 100 samples per group (less than the 112 mentioned in the previous publication) estimated with this tool is similar to the power estimated by the previous one with a mean difference of 10%, with almost all the probes (99%) showing power higher than 80% (see results below).

3. Unfortunately, I think the authors have misinterpreted this – as just using the genes covered by the probes as background does not take into consideration that some Cp genes have disproportionally more probes. Please refer to Maksimovic et al. (2021) Genome Biology for non-biased analysis.

Good point again, sorry for the misunderstanding. But yes, differential gene-CpG coverage and the fact that each probe could be annotated to different genes might cause undesirable biases in the enrichment analysis. We were not aware of *gsameth* function, so thanks for pointing this out, which for sure we will use in future EWAS.

In order to check if these biases influenced our gene based enrichment analyses, we repeated the enrichment analysis with *gsameth* function from *missMethyl* R package (which takes into consideration both potential types of biases). As you can see in the results below, the main pathways were found to be enriched also with the *gsameth* function, including the interferon signaling, CD209 signaling, FCGR3A-mediated phagocytosis and the potential therapeutics for SARS in the S.He module. These analyses are now included as supplementary figures 3 and 7 in the manuscript, mentioned in the results and methods sections, and *missMethyl* and Maksimovic et al. 2021 references added to the manuscript.